# Efficient Minimax Strategies for Square Loss Games

**Wouter M. Koolen**
Queensland University of Technology and UC Berkeley
wouter.koolen@qut.edu.au

**Alan Malek**
University of California, Berkeley
malek@eecs.berkeley.edu

**Peter L. Bartlett**
University of California, Berkeley and Queensland University of Technology
peter@berkeley.edu

## Abstract

We consider online prediction problems where the loss between the prediction and the outcome is measured by the squared Euclidean distance and its generalization, the squared Mahalanobis distance. We derive the minimax solutions for the case where the prediction and action spaces are the simplex (this setup is sometimes called the Brier game) and the $\ell_2$ ball (this setup is related to Gaussian density estimation). We show that in both cases the value of each sub-game is a quadratic function of a simple statistic of the state, with coefficients that can be efficiently computed using an explicit recurrence relation. The resulting deterministic minimax strategy and randomized maximin strategy are linear functions of the statistic.

## 1 Introduction

We are interested in general strategies for sequential prediction and decision making (a.k.a. online learning) that improve their performance with experience. Since the early days of online learning, people have formalized such learning tasks as regret games. The learner interacts with an adversarial environment with the goal of performing almost as well as the best strategy from some fixed reference set. In many cases, we have efficient algorithms with an upper bound on the regret that meets the game-theoretic lower bound (up to a small constant factor). In a few special cases, we have the exact minimax strategy, meaning that we understand the learning problem at all levels of detail. In even fewer cases we can also efficiently execute the minimax strategy. These cases serve as exemplars to guide our thinking about learning algorithms.

In this paper we add two interesting examples to the canon of efficiently computable minimax strategies. Our setup, as described in Figure 1, is as follows. The Learner and the Adversary play vectors $\boldsymbol{a} \in \mathcal{A}$ and $\boldsymbol{x} \in \mathcal{X}$, upon which the Learner is penalized using the squared Euclidean distance $\|\boldsymbol{a} - \boldsymbol{x}\|^2$ or its generalization, the squared Mahalanobis distance,

$$\|\boldsymbol{a} - \boldsymbol{x}\|_{\boldsymbol{W}}^2 \; = \; (\boldsymbol{a} - \boldsymbol{x})^{\intercal} \boldsymbol{W}^{-1} (\boldsymbol{a} - \boldsymbol{x}),$$

parametrized by a symmetric matrix $\boldsymbol{W} \succ \boldsymbol{0}$. After a sequence of $T$ such interactions, we compare the loss of the Learner to the loss of the best fixed prediction $\boldsymbol{a}^* \in \mathcal{A}$. In all our examples, this best fixed action in hindsight is the mean outcome $\boldsymbol{a}^* = \frac{1}{T} \sum_{t=1}^{T} \boldsymbol{x}_t$, regardless of $\boldsymbol{W}$. We use regret, the difference between the loss of the learner and the loss of $\boldsymbol{a}^*$, to evaluate performance. The *minimax regret* for the $T$-round game, also known as the value of the game, is given by

$$V \; := \; \inf_{\boldsymbol{a}_1} \sup_{\boldsymbol{x}_1} \cdots \inf_{\boldsymbol{a}_T} \sup_{\boldsymbol{x}_T} \sum_{t=1}^{T} \frac{1}{2} \|\boldsymbol{a}_t - \boldsymbol{x}_t\|_{\boldsymbol{W}}^2 - \inf_{\boldsymbol{a}} \sum_{t=1}^{T} \frac{1}{2} \|\boldsymbol{a} - \boldsymbol{x}_t\|_{\boldsymbol{W}}^2 \tag{1}$$

where the $\boldsymbol{a}_t$ range over actions $\mathcal{A}$ and the $\boldsymbol{x}_t$ range over outcomes $\mathcal{X}$. The *minimax strategy* chooses the $\boldsymbol{a}_t$, given all past outcomes $\boldsymbol{x}_1, \ldots, \boldsymbol{x}_{t-1}$, to achieve this regret. Intuitively, the minimax regret is the regret if both players play optimally while assuming the other player is doing the same.

Our first example is the Brier game, where the action and outcome spaces are the probability simplex with $K$ outcomes. The Brier game is traditionally popular in meteorology [Bri50].

Our second example is the ball game, where the action and outcome spaces are the Euclidean norm ball, i.e. $\mathcal{A} = \mathcal{X} = \{\boldsymbol{x} \in \mathbb{R}^K \mid \|\boldsymbol{x}\|_2 = 1\}$. (Even though we measure loss by the squared Mahalanobis distance, we play on the standard Euclidean norm ball.) The ball game is related to Gaussian density estimation [TW00].

Given: $T, \boldsymbol{W}, \mathcal{A}, \mathcal{X}$.
For $t = 1, 2, \ldots, T$

- Learner chooses prediction $\boldsymbol{a}_t \in \mathcal{A}$
- Adversary chooses outcome $\boldsymbol{x}_t \in \mathcal{X}$
- Learner incurs loss $\frac{1}{2}\|\boldsymbol{a}_t - \boldsymbol{x}_t\|_{\boldsymbol{W}}^2$.

Figure 1: Protocol

In each case we exhibit a strategy that can play a $T$-round game in $O(TK^2)$ time. (The algorithm spends $O(TK + K^3)$ time pre-processing the game, and then plays in $O(K^2)$ time per round.)

## 2 Outline

We define our loss using the squared Mahalanobis distance, parametrized by a symmetric matrix $\boldsymbol{W} \succ \boldsymbol{0}$. We recover the squared Euclidean distance by choosing $\boldsymbol{W} = \boldsymbol{I}$. Our games will always last $T$ rounds. For some observed data $\boldsymbol{x}_1, \ldots, \boldsymbol{x}_n$, the *value-to-go* for the remaining $T - n$ rounds is given by

$$V(\boldsymbol{x}_1, \ldots, \boldsymbol{x}_n) \;:=\; \inf_{\boldsymbol{a}_{n+1}} \sup_{\boldsymbol{x}_{n+1}} \cdots \inf_{\boldsymbol{a}_T} \sup_{\boldsymbol{x}_T} \sum_{t=n+1}^{T} \frac{1}{2}\|\boldsymbol{a}_t - \boldsymbol{x}_t\|_{\boldsymbol{W}}^2 - \inf_{\boldsymbol{a}} \sum_{t=1}^{T} \frac{1}{2}\|\boldsymbol{a} - \boldsymbol{x}_t\|_{\boldsymbol{W}}^2.$$

By definition, the minimax regret (1) is $V = V(\epsilon)$ where $\epsilon$ is the empty sequence, and the value-to-go satisfies the recurrence

$$V(\boldsymbol{x}_1, \ldots, \boldsymbol{x}_n) \;=\; \begin{cases} -\inf_{\boldsymbol{a}} \sum_{t=1}^{T} \frac{1}{2}\|\boldsymbol{a} - \boldsymbol{x}_t\|_{\boldsymbol{W}}^2 & \text{if } n = T, \\ \inf_{\boldsymbol{a}_{n+1}} \sup_{\boldsymbol{x}_{n+1}} \frac{1}{2}\|\boldsymbol{a}_{n+1} - \boldsymbol{x}_{n+1}\|_{\boldsymbol{W}}^2 + V(\boldsymbol{x}_1, \ldots, \boldsymbol{x}_{n+1}) & \text{if } n < T. \end{cases} \quad (2)$$

Our analysis for the two games proceeds in a similar manner. For some past history of plays $(\boldsymbol{x}_1, \ldots, \boldsymbol{x}_n)$ of length $n$, we summarize the state by $\boldsymbol{s} = \sum_{t=1}^{n} \boldsymbol{x}_t$ and $\sigma^2 = \sum_{t=1}^{n} \boldsymbol{x}_t^\mathsf{T} \boldsymbol{W}^{-1} \boldsymbol{x}_t$. As we will see, the value-to-go after $n$ of $T$ rounds can be written as $V(\boldsymbol{s}, \sigma^2, n)$; i.e. it only depends on the past plays through $\boldsymbol{s}$ and $\sigma^2$. More surprisingly, for each $n$, the value-to-go $V(\boldsymbol{s}, \sigma^2, n)$ is a quadratic function of $\boldsymbol{s}$ and a linear function of $\sigma^2$ (under certain conditions on $\boldsymbol{W}$). While it is straightforward to see that the terminal value $V(\boldsymbol{s}, \sigma^2, T)$ is quadratic in the state (this is easily checked by computing the loss of the best expert and using the first case of Equation (2)), it is not at all obvious that propagating from $V(\boldsymbol{s} + \boldsymbol{x}, \sigma^2 + \boldsymbol{x}^\mathsf{T} \boldsymbol{W}^{-1} \boldsymbol{x}, n+1)$ to $V(\boldsymbol{s}, \sigma^2, n)$, using the second case of (2), preserves this structure.

This compact representation of the value-function is an essential ingredient for a computationally feasible algorithm. Many minimax approaches, such as normalized maximum likelihood [Sht87], have computational complexities that scale exponentially with the time horizon. We derive a strategy that can play in constant amortized time.

Why is this interesting? We go beyond previous work in a few directions. First, we exhibit two new games that belong to the tiny class admitting computationally feasible minimax algorithms. Second, we consider the setting with squared Mahalanobis loss which allows the user intricate control over the penalization of different prediction errors. Our results clearly show how the learner should exploit this prioritization.

### 2.1 Related work

Repeated games with minimax strategies are frequently studied ([CBL06]) and, in online learning, minimax analysis has been applied to a variety of losses and repeated games; however, computa-

tionally feasible algorithms are the exception, not the rule. For example, consider log loss, first discussed in [Sht87]. Whiile the minimax algorithm, Normalized Maximum Likelihood, is well known [CBL06], it generally requires computation that is exponential in the time horizon as one needs to aggregate over all data sequences. To our knowledge, there are two exceptions where efficient NML forecasters are possible: the multinomial case where fast Fourier transforms may be exploited [KM05], and very particular exponential families that cause NML to be a Bayesian strategy [HB12], [BGH$^+$13]. The minimax optimal strategy is known also for: (i) the ball game with $W = I$ [TW00] (our generalization to Mahalanobis $W \neq I$ results in fundamentally different strategies), (ii) the ball game with $W = I$ and a constraint on the player's deviation from the current empirical minimizer [ABRT08] (for which the optimal strategy is Follow-the-Leader), (iii) Lipschitz-bounded convex loss functions [ABRT08], (iv) experts with an $L^*$ bound [AWY08], and (v) static experts with absolute loss [CBS11]. While not guaranteed to be an exhaustive list, the previous paragraph demonstrates the rarity of tractable minimax algorithms.

## 3 The Offline Problem

The regret is defined as the difference between the loss of the algorithm and the loss of the best action in hindsight. Here we calculate that action and its loss.

**Lemma 3.1.** *Suppose $\mathcal{A} \supseteq \mathrm{conv}(\mathcal{X})$ (this will always hold in the settings we study). For data $\boldsymbol{x}_1, \ldots, \boldsymbol{x}_T \in \mathcal{X}$, the loss of the best action in hindsight equals*

$$\inf_{\boldsymbol{a} \in \mathcal{A}} \sum_{t=1}^{T} \frac{1}{2} \|\boldsymbol{a} - \boldsymbol{x}_t\|_{\boldsymbol{W}}^2 = \frac{1}{2} \left( \sum_{t=1}^{T} \boldsymbol{x}_t^\mathsf{T} \boldsymbol{W}^{-1} \boldsymbol{x}_t - \frac{1}{T} \left( \sum_{t=1}^{T} \boldsymbol{x}_t \right)^\mathsf{T} \boldsymbol{W}^{-1} \left( \sum_{t=1}^{T} \boldsymbol{x}_t \right) \right), \quad (3)$$

*and the minimizer is the mean outcome $\boldsymbol{a}^* = \frac{1}{T} \sum_{t=1}^{T} \boldsymbol{x}_t$.*

*Proof.* The unconstrained minimizer and value are obtained by equating the derivative to zero and plugging in the solution. The assumption $\mathcal{A} \supseteq \mathrm{conv}(\mathcal{X})$ ensures that the constraint $\boldsymbol{a} \in \mathcal{A}$ is inactive. $\square$

The best action in hindsight is curiously independent of $\boldsymbol{W}$, $\mathcal{A}$ and $\mathcal{X}$. This also shows that the follow the leader strategy that plays $\boldsymbol{a}_t = \frac{1}{t-1} \sum_{s=1}^{t-1} \boldsymbol{x}_s$ is independent of $\boldsymbol{W}$ and $\mathcal{A}$ as well. As we shall see, the minimax strategy does not have this property.

## 4 Simplex (Brier) Game

In this section we analyze the Brier game. The action and outcome spaces are the probability simplex on $K$ outcomes; $\mathcal{A} = \mathcal{X} = \triangle := \{\boldsymbol{x} \in \mathbb{R}_+^K \mid \mathbf{1}^\mathsf{T}\boldsymbol{x} = 1\}$. The loss is given by half the squared Mahalanobis distance, $\frac{1}{2}\|\boldsymbol{a} - \boldsymbol{x}\|_{\boldsymbol{W}}^2$. We present a full minimax analysis of the $T$-round game: we calculate the game value, derive the maximin and minimax strategies, and discuss their efficient implementation.

The structure of this section is as follows. In Lemmas 4.1 and 4.2, the conclusions (value and optimizers) are obtained under the proviso that the given optimizer lies in the simplex. In our main result, Theorem 4.3, we apply these auxiliary results to our minimax analysis and argue that the maximizer indeed lies in the simplex. We immediately work from a general symmetric $\boldsymbol{W} \succ \boldsymbol{0}$ with the following lemma.

**Lemma 4.1.** *Fix a symmetric matrix $\boldsymbol{C} \succ \boldsymbol{0}$ and vector $\boldsymbol{d}$. The optimization problem*

$$\max_{\boldsymbol{p} \in \triangle} -\frac{1}{2}\boldsymbol{p}^\mathsf{T}\boldsymbol{C}^{-1}\boldsymbol{p} + \boldsymbol{d}^\mathsf{T}\boldsymbol{p}$$

*has value $\frac{1}{2}\left(\boldsymbol{d}^\mathsf{T}\boldsymbol{C}\boldsymbol{d} - \frac{(\mathbf{1}^\mathsf{T}\boldsymbol{C}\boldsymbol{d}-1)^2}{\mathbf{1}^\mathsf{T}\boldsymbol{C}\mathbf{1}}\right) = \frac{1}{2}\left(\boldsymbol{d}^\mathsf{T}\left(\boldsymbol{C} - \frac{\boldsymbol{C}\mathbf{1}\mathbf{1}^\mathsf{T}\boldsymbol{C}}{\mathbf{1}^\mathsf{T}\boldsymbol{C}\mathbf{1}}\right)\boldsymbol{d} + \frac{2\mathbf{1}^\mathsf{T}\boldsymbol{C}\boldsymbol{d}-1}{\mathbf{1}^\mathsf{T}\boldsymbol{C}\mathbf{1}}\right)$ attained at optimizer*

$$\boldsymbol{p}^* = \boldsymbol{C}\left(\boldsymbol{d} - \frac{\mathbf{1}^\mathsf{T}\boldsymbol{C}\boldsymbol{d}-1}{\mathbf{1}^\mathsf{T}\boldsymbol{C}\mathbf{1}}\mathbf{1}\right) = \left(\boldsymbol{C} - \frac{\boldsymbol{C}\mathbf{1}\mathbf{1}^\mathsf{T}\boldsymbol{C}}{\mathbf{1}^\mathsf{T}\boldsymbol{C}\mathbf{1}}\right)\boldsymbol{d} + \frac{\boldsymbol{C}\mathbf{1}}{\mathbf{1}^\mathsf{T}\boldsymbol{C}\mathbf{1}}$$

*provided that $\boldsymbol{p}^*$ is in the simplex.*

*Proof.* We solve for the optimal $\boldsymbol{p}^*$. Introducing Lagrange multiplier $\lambda$ for the constraint $\sum_k p_k = 1$, we need to have $\boldsymbol{p} = \boldsymbol{C}\,(\boldsymbol{d} - \lambda\mathbf{1})$ which results in $\lambda = \frac{\mathbf{1}^\mathsf{T} \boldsymbol{C} \boldsymbol{d} - 1}{\mathbf{1}^\mathsf{T} \boldsymbol{C} \mathbf{1}}$. Thus, the maximizer equals $\boldsymbol{p}^* = \boldsymbol{C}\left(\boldsymbol{d} - \frac{\mathbf{1}^\mathsf{T} \boldsymbol{C} \boldsymbol{d} - 1}{\mathbf{1}^\mathsf{T} \boldsymbol{C} \mathbf{1}}\mathbf{1}\right)$ which produces objective value $\frac{1}{2}\left(\boldsymbol{d} + \frac{\mathbf{1}^\mathsf{T} \boldsymbol{C} \boldsymbol{d} - 1}{\mathbf{1}^\mathsf{T} \boldsymbol{C} \mathbf{1}}\mathbf{1}\right)^\mathsf{T} \boldsymbol{C}\left(\boldsymbol{d} - \frac{\mathbf{1}^\mathsf{T} \boldsymbol{C} \boldsymbol{d} - 1}{\mathbf{1}^\mathsf{T} \boldsymbol{C} \mathbf{1}}\mathbf{1}\right)$. The statement follows from simplification. $\qquad\square$

This lemma allows us to compute the value and saddle point whenever the future payoff is quadratic.

**Lemma 4.2.** *Fix symmetric matrices $\boldsymbol{W} \succeq \boldsymbol{0}$ and $\boldsymbol{A}$ such that $\boldsymbol{W}^{-1} + \boldsymbol{A} \succeq \boldsymbol{0}$, and a vector $\boldsymbol{b}$. The optimization problem*

$$\min_{\boldsymbol{a}\in\triangle}\max_{\boldsymbol{x}\in\triangle} \frac{1}{2}\|\boldsymbol{a} - \boldsymbol{x}\|_{\boldsymbol{W}}^2 + \frac{1}{2}\boldsymbol{x}^\mathsf{T}\boldsymbol{A}\boldsymbol{x} + \boldsymbol{b}^\mathsf{T}\boldsymbol{x}$$

*achieves its value*

$$\frac{1}{2}\boldsymbol{c}^\mathsf{T}\boldsymbol{W}\boldsymbol{c} - \frac{1}{2}\frac{(\mathbf{1}^\mathsf{T}\boldsymbol{W}\boldsymbol{c} - 1)^2}{\mathbf{1}^\mathsf{T}\boldsymbol{W}\mathbf{1}} \qquad \text{where} \qquad \boldsymbol{c} = \frac{1}{2}\operatorname{diag}\left(\boldsymbol{W}^{-1} + \boldsymbol{A}\right) + \boldsymbol{b}$$

*at saddle point (the maximin strategy randomizes, playing $\boldsymbol{x} = \boldsymbol{e}_i$ with probability $p_i^*$)*

$$\boldsymbol{a}^* = \boldsymbol{p}^* = \left(\boldsymbol{W} - \frac{\boldsymbol{W}\mathbf{1}\mathbf{1}^\mathsf{T}\boldsymbol{W}}{\mathbf{1}^\mathsf{T}\boldsymbol{W}\mathbf{1}}\right)\boldsymbol{c} + \frac{\boldsymbol{W}\mathbf{1}}{\mathbf{1}^\mathsf{T}\boldsymbol{W}\mathbf{1}}$$

*provided $\boldsymbol{p}^* \succeq \boldsymbol{0}$.*

*Proof.* The objective is convex in $\boldsymbol{x}$ for each $\boldsymbol{a}$ as $\boldsymbol{W}^{-1} + \boldsymbol{A} \succeq \boldsymbol{0}$, so it is maximized at a corner $\boldsymbol{x} = \boldsymbol{e}_k$. We apply min-max swap (see e.g. [Sio58]), properness of the loss (which implies that $\boldsymbol{a}^* = \boldsymbol{p}^*$) and expand:

$$
\begin{aligned}
&\min_{\boldsymbol{a}\in\triangle}\max_{\boldsymbol{x}\in\triangle} \frac{1}{2}\|\boldsymbol{a} - \boldsymbol{x}\|_{\boldsymbol{W}}^2 + \frac{1}{2}\boldsymbol{x}^\mathsf{T}\boldsymbol{A}\boldsymbol{x} + \boldsymbol{b}^\mathsf{T}\boldsymbol{x} \\
&= \min_{\boldsymbol{a}\in\triangle}\max_{k} \frac{1}{2}\|\boldsymbol{a} - \boldsymbol{e}_k\|_{\boldsymbol{W}}^2 + \frac{1}{2}\boldsymbol{e}_k^\mathsf{T}\boldsymbol{A}\boldsymbol{e}_k + \boldsymbol{b}^\mathsf{T}\boldsymbol{e}_k \\
&= \max_{\boldsymbol{p}\in\triangle}\min_{\boldsymbol{a}\in\triangle}\mathbb{E}_{k\sim\boldsymbol{p}}\left[\frac{1}{2}\|\boldsymbol{a} - \boldsymbol{e}_k\|_{\boldsymbol{W}}^2 + \frac{1}{2}\boldsymbol{e}_k^\mathsf{T}\boldsymbol{A}\boldsymbol{e}_k + \boldsymbol{b}^\mathsf{T}\boldsymbol{e}_k\right] \\
&= \max_{\boldsymbol{p}\in\triangle}\mathbb{E}_{k\sim\boldsymbol{p}}\left[\frac{1}{2}\|\boldsymbol{p} - \boldsymbol{e}_k\|_{\boldsymbol{W}}^2 + \frac{1}{2}\boldsymbol{e}_k^\mathsf{T}\boldsymbol{A}\boldsymbol{e}_k + \boldsymbol{b}^\mathsf{T}\boldsymbol{e}_k\right] \\
&= \max_{\boldsymbol{p}\in\triangle} -\frac{1}{2}\boldsymbol{p}^\mathsf{T}\boldsymbol{W}^{-1}\boldsymbol{p} + \frac{1}{2}\operatorname{diag}\left(\boldsymbol{W}^{-1} + \boldsymbol{A}\right)^\mathsf{T}\boldsymbol{p} + \boldsymbol{b}^\mathsf{T}\boldsymbol{p}
\end{aligned}
$$

The proof is completed by applying Lemma 4.1. $\qquad\square$

## 4.1 Minimax Analysis of the Brier Game

Next, we turn to computing $V(\boldsymbol{s}, \sigma^2, n)$ as a recursion and specifying the minimax and maximin strategies. However, for the value-to-go function to retain its quadratic form, we need an alignment condition on $\boldsymbol{W}$. We say that $\boldsymbol{W}$ is *aligned with the simplex* if

$$\left(\boldsymbol{W} - \frac{\boldsymbol{W}\mathbf{1}\mathbf{1}^\mathsf{T}\boldsymbol{W}}{\mathbf{1}^\mathsf{T}\boldsymbol{W}\mathbf{1}}\right)\operatorname{diag}(\boldsymbol{W}^{-1}) \succeq -2\frac{\boldsymbol{W}\mathbf{1}}{\mathbf{1}^\mathsf{T}\boldsymbol{W}\mathbf{1}}, \tag{4}$$

where $\succeq$ denotes an entry-wise inequality between vectors. Note that many matrices besides $\boldsymbol{I}$ satisfy this condition: for example, all symmetric $2 \times 2$ matrices. We can now fully specify the value and strategies for the Brier game.

**Theorem 4.3.** *Consider the $T$-round Brier game with Mahalanobis loss $\frac{1}{2}\|\boldsymbol{a} - \boldsymbol{x}\|_{\boldsymbol{W}}^2$ with $\boldsymbol{W}$ satisfying the alignment condition (4). After $n$ outcomes $(\boldsymbol{x}_1, \ldots, \boldsymbol{x}_n)$ with statistics $\boldsymbol{s} = \sum_{t=1}^n \boldsymbol{x}_t$ and $\sigma^2 = \sum_{t=1}^n \boldsymbol{x}_t^\mathsf{T}\boldsymbol{W}^{-1}\boldsymbol{x}_t$ the value-to-go is*

$$V(\boldsymbol{s}, \sigma^2, n) = \frac{1}{2}\alpha_n \boldsymbol{s}^\mathsf{T}\boldsymbol{W}^{-1}\boldsymbol{s} - \frac{1}{2}\sigma^2 + \frac{1}{2}(1 - n\alpha_n)\operatorname{diag}(\boldsymbol{W}^{-1})^\mathsf{T}\boldsymbol{s} + \gamma_n,$$

*and the minimax and maximin strategies are given by*

$$\boldsymbol{a}^*(\boldsymbol{s}, \sigma^2, n) = \boldsymbol{p}^*(\boldsymbol{s}, \sigma^2, n) = \frac{\boldsymbol{W}\mathbf{1}}{\mathbf{1}^\mathsf{T}\boldsymbol{W}\mathbf{1}} + \alpha_{n+1}\left(\boldsymbol{s} - \frac{n\boldsymbol{W}\mathbf{1}}{\mathbf{1}^\mathsf{T}\boldsymbol{W}\mathbf{1}}\right)$$

$$+ \frac{1}{2}(1 - n\alpha_{n+1})\left(\boldsymbol{W} - \frac{\boldsymbol{W}\mathbf{1}\mathbf{1}^\mathsf{T}\boldsymbol{W}}{\mathbf{1}^\mathsf{T}\boldsymbol{W}\mathbf{1}}\right)\operatorname{diag}(\boldsymbol{W}^{-1})$$

*where the coefficients are defined recursively by*

$$\alpha_T = \frac{1}{T} \qquad\qquad \gamma_T = 0$$

$$\alpha_n = \alpha_{n+1}^2 + \alpha_{n+1}$$

$$\gamma_n = \frac{(1 - n\alpha_{n+1})^2}{2}\left(\frac{1}{4}\operatorname{diag}(\boldsymbol{W}^{-1})^\mathsf{T}\boldsymbol{W}\operatorname{diag}(\boldsymbol{W}^{-1}) - \frac{\left(\frac{1}{2}\mathbf{1}^\mathsf{T}\boldsymbol{W}\operatorname{diag}(\boldsymbol{W}^{-1}) - 1\right)^2}{\mathbf{1}^\mathsf{T}\boldsymbol{W}\mathbf{1}}\right)$$

$$+ \gamma_{n+1}.$$

*Proof.* We prove this by induction, beginning at the end of the game and working backwards in time. Assume that $V(\boldsymbol{s}, \sigma^2, T)$ has the given form. Recall that the value at the end of the game is $V(\boldsymbol{s}, \sigma^2, T) = -\inf_{\boldsymbol{a}}\sum_{t=1}^T \frac{1}{2}\|\boldsymbol{a} - \boldsymbol{x}_t\|_{\boldsymbol{W}}^2$ and is given by Lemma 3.1. Matching coefficients, we find $V(\boldsymbol{s}, \sigma^2, T)$ corresponds to $\alpha_T = \frac{1}{T}$ and $\gamma_T = 0$.

Now assume that $V$ has the assumed form after $n$ rounds. Using $\boldsymbol{s}$ and $\sigma^2$ to denote the state after $n - 1$ rounds, we can write

$$V(\boldsymbol{s}, \sigma^2, n - 1) = \min_{\boldsymbol{a}\in\triangle}\max_{\boldsymbol{x}\in\triangle}\frac{1}{2}\|\boldsymbol{a} - \boldsymbol{x}\|_{\boldsymbol{W}}^2 + \frac{1}{2}\alpha_n(\boldsymbol{s} + \boldsymbol{x})^\mathsf{T}\boldsymbol{W}^{-1}(\boldsymbol{s} + \boldsymbol{x})$$

$$- \frac{1}{2}(\sigma^2 + \boldsymbol{x}^\mathsf{T}\boldsymbol{W}^{-1}\boldsymbol{x}) + \frac{1}{2}(1 - n\alpha_n)\operatorname{diag}(\boldsymbol{W}^{-1})^\mathsf{T}(\boldsymbol{s} + \boldsymbol{x}) + \gamma_n.$$

Using Lemma 4.2 to evaluate the right hand side produces a quadratic function in the state, and we can then match terms to find $\alpha_{n-1}$ and $\gamma_{n-1}$ and the minimax and maximin strategy. The final step is checking the $\boldsymbol{p}^* \succeq 0$ condition necessary to apply Lemma 4.2, which is equivalent to $\boldsymbol{W}$ being aligned with the simplex. See the appendix for a complete proof. □

This full characterization of the game allows us to derive the following minimax regret bound.

**Theorem 4.4.** *Let $\boldsymbol{W}$ satisfy the alignment condition* (4)*. The minimax regret of the $T$-round simplex game satisfies*

$$V \leq \frac{1 + \ln(T)}{2}\left(\frac{1}{4}\operatorname{diag}(\boldsymbol{W}^{-1})^\mathsf{T}\boldsymbol{W}\operatorname{diag}(\boldsymbol{W}^{-1}) - \frac{\left(\frac{1}{2}\mathbf{1}^\mathsf{T}\boldsymbol{W}\operatorname{diag}(\boldsymbol{W}^{-1}) - 1\right)^2}{\mathbf{1}^\mathsf{T}\boldsymbol{W}\mathbf{1}}\right).$$

*Proof.* The regret is equal to the value of the game, $V = V(\boldsymbol{0}, 0, 0) = \gamma_0$. First observe that

$$(1 - n\alpha_{n+1})^2 = 1 - 2n\alpha_{n+1} + n^2\alpha_{n+1}^2$$

$$= 1 - 2n\alpha_{n+1} + n^2(\alpha_n - \alpha_{n+1})$$

$$= \alpha_{n+1} + 1 - (n + 1)^2\alpha_{n+1} + n^2\alpha_n.$$

After summing over $n$ the last two terms telescope, and we find

$$\gamma_0 \propto \sum_{n=0}^{T-1}(1 - n\alpha_{n+1})^2 = -T^2\alpha_T + \sum_{n=0}^{T-1}(1 + \alpha_{n+1}) = \sum_{n=1}^T \alpha_n.$$

Each $\alpha_n$ can be bounded by $1/n$, as observed in [TW00, proof of Lemma 2]. In the base case $n = T$ this holds with equality, and for $n < T$ we have

$$\alpha_n = \alpha_{n+1}^2 + \alpha_{n+1} \leq \frac{1}{(n+1)^2} + \frac{1}{n+1} = \frac{1}{n}\frac{n(n+2)}{(n+1)^2} \leq \frac{1}{n}.$$

It follows that $\gamma_0 \propto \sum_{n=1}^T \alpha_n \leq \sum_{n=1}^T \frac{1}{n} \leq 1 + \ln(T)$ as desired. □

# 5  Norm Ball Game

This section parallels the previous. Here, we consider the online game with Mahalanobis loss and $\mathcal{A} = \mathcal{X} = \bigcirc := \{ x \in \mathbb{R}^K \mid \|x\| \leq 1 \}$, the 2-norm Euclidian ball (not the Mahalanobis ball). We show that the value-to-go function is always quadratic in $s$ and linear in $\sigma^2$ and derive the minimax and maximin strategies.

**Lemma 5.1.** *Fix a symmetric matrix $A$ and vector $b$ and assume $A + W^{-1} \succ 0$. Let $\lambda_{\max}$ be the largest eigenvalue of $W^{-1} + A$ and $v_{\max}$ the corresponding eigenvector. If $b^{\intercal} \left( \lambda_{\max} I - A \right)^{-2} b \leq 1$, then the optimization problem*

$$\inf_{a \in \bigcirc} \sup_{x \in \bigcirc} \frac{1}{2} \|a - x\|_W^2 + \frac{1}{2} x^{\intercal} A x + x^{\intercal} b$$

*has value $\frac{1}{2} b^{\intercal} \left( \lambda_{\max} I - A \right)^{-1} b + \frac{1}{2} \lambda_{\max}$, minimax strategy $a^* = (\lambda_{\max} I - A)^{-1} b$ and a randomized maximin strategy that plays two unit length vectors, with*

$$\Pr \left( x = a_{\perp} \pm \sqrt{1 - a_{\perp}^{\intercal} a_{\perp}} \, v_{\max} \right) = \frac{1}{2} \pm \frac{1}{2} \sqrt{\frac{a_{\parallel}^{\intercal} a_{\parallel}}{1 - a_{\perp}^{\intercal} a_{\perp}}},$$

*where $a_{\perp}$ and $a_{\parallel}$ are the components of $a^*$ perpendicular and parallel to $v_{\max}$.*

*Proof.* As the objective is convex, the inner optimum must be on the boundary and hence will be at a unit vector $x$. Introduce a Lagrange multiplier $\lambda$ for $x^{\intercal} x \leq 1$ to get the Lagrangian

$$\inf_{a \in \bigcirc} \inf_{\lambda \geq 0} \sup_{x} \frac{1}{2} \|a - x\|_W^2 + \frac{1}{2} x^{\intercal} A x + x^{\intercal} b + \lambda \frac{1}{2} (1 - x^{\intercal} x).$$

This is concave in $x$ if $W^{-1} + A - \lambda I \preceq 0$, that is, $\lambda_{\max} \leq \lambda$. Differentiating yields the optimizer $x^* = (W^{-1} + A - \lambda I)^{-1}(W^{-1} a - b)$, which leaves us with an optimization in only $a$ and $\lambda$:

$$\inf_{a \in \bigcirc} \inf_{\lambda \geq \lambda_{\max}} \frac{1}{2} a^{\intercal} W^{-1} a - \frac{1}{2} (W^{-1} a - b)^{\intercal} (W^{-1} + A - \lambda I)^{-1} (W^{-1} a - b) + \frac{1}{2} \lambda.$$

Since the infimums are over closed sets, we can exchange their order. Unconstrained optimization of $a$ results in $a^* = (\lambda I - A)^{-1} b$. Evaluating the objective at $a^*$ and using $W^{-1} a^* - b = W^{-1} (\lambda I - A)^{-1} b - b = (W^{-1} + A - \lambda I)(\lambda I - A)^{-1} b$ results in

$$\inf_{\lambda \geq \lambda_{\max}} \frac{1}{2} b^{\intercal} (\lambda I - A)^{-1} b + \frac{1}{2} \lambda = \inf_{\lambda \geq \lambda_{\max}} \frac{1}{2} \left( \sum_i \frac{(u_i^{\intercal} b)^2}{\lambda - \lambda_i} + \lambda \right),$$

using the spectral decomposition $A = \sum_i \lambda_i u_i u_i^{\intercal}$. For $\lambda \geq \lambda_{\max}$, we have $\lambda \geq \lambda_i$. Taking derivatives, provided $b^{\intercal} \left( \lambda_{\max} I - A \right)^{-2} b \leq 1$, this function is increasing in $\lambda \geq \lambda_{\max}$, and so obtains its infimum at $\lambda_{\max}$. Thus, when the assumed inequality is satisfied, the $a^*$ is minimax for the given $x^*$.

To obtain the maximin strategy, we can take the usual convexification where the Adversary plays distributions $P$ over the unit sphere. This allows us to swap the infimum and supremum (see e.g. Sion's minimax theorem[Sio58]) and obtain an equivalent optimization problem. We then see that the objective only depends on the mean $\mu = \mathbb{E} \, x$ and second moment $D = \mathbb{E} \, x x^{\intercal}$ of the distribution $P$. The characterization in [KNW13, Theorem 2.1] tells us that $\mu, D$ are the first two moments of a distribution on units iff $\operatorname{tr}(D) = 1$ and $D \succeq \mu \mu^{\intercal}$. Then, our usual min-max swap yields

$$V = \sup_P \inf_{a \in \bigcirc} \mathbb{E}_{x \sim P} \left[ \frac{1}{2} a^{\intercal} W^{-1} a - a^{\intercal} W^{-1} x + \frac{1}{2} x^{\intercal} W^{-1} x + \frac{1}{2} x^{\intercal} A x + b^{\intercal} x \right]$$

$$= \sup_{\mu, D} \inf_{a \in \bigcirc} \frac{1}{2} a^{\intercal} W^{-1} a - a^{\intercal} W^{-1} \mu + \frac{1}{2} \operatorname{tr} \left( (W^{-1} + A) D \right) + b^{\intercal} \mu$$

$$= \sup_{\mu, D} -\frac{1}{2} \mu^{\intercal} W^{-1} \mu + \frac{1}{2} \operatorname{tr} \left( (W^{-1} + A) D \right) + b^{\intercal} \mu$$

$$= -\frac{1}{2} a^{*\intercal} W^{-1} a^* + b^{\intercal} a^* + \sup_{\substack{D \succeq a^* a^{*\intercal} \\ \operatorname{tr}(D) = 1}} \frac{1}{2} \operatorname{tr} \left( (W^{-1} + A) D \right)$$

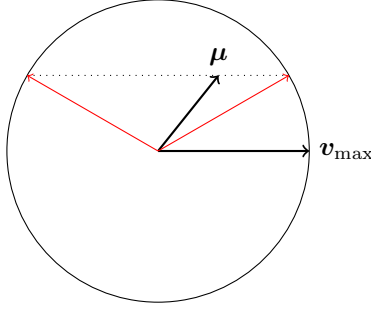

Figure 2: Illustration of the maximin distribution from Lemma 5.1. The mixture of red unit vectors with mean $\boldsymbol{\mu}$ has second moment $\boldsymbol{D} = \boldsymbol{\mu}\boldsymbol{\mu}^\mathsf{T} + (1 - \boldsymbol{\mu}^\mathsf{T}\boldsymbol{\mu})\boldsymbol{v}_{\max}\boldsymbol{v}_{\max}^\mathsf{T}$.

where the second equality uses $\boldsymbol{a} = \boldsymbol{\mu}$ and the third used the saddle point condition $\boldsymbol{\mu}^* = \boldsymbol{a}^*$. The matrix $\boldsymbol{D}$ with constraint $\mathrm{tr}(\boldsymbol{D}) = 1$ now seeks to align with the largest eigenvector of $\boldsymbol{W}^{-1} + \boldsymbol{A}$ but it also has to respect the constraint $\boldsymbol{D} \succeq \boldsymbol{a}^*\boldsymbol{a}^{*\mathsf{T}}$. We now re-parameterise by $\boldsymbol{C} = \boldsymbol{D} - \boldsymbol{a}^*\boldsymbol{a}^{*\mathsf{T}}$. We then need to find

$$\sup_{\substack{\boldsymbol{C} \succeq \boldsymbol{0} \\ \mathrm{tr}(\boldsymbol{C}) = 1 - \boldsymbol{a}^{*\mathsf{T}}\boldsymbol{a}^*}} \frac{1}{2}\,\mathrm{tr}\left((\boldsymbol{W}^{-1} + \boldsymbol{A})\boldsymbol{C}\right).$$

By linearity of the objective the maximizer is of rank 1, and hence this is a (scaled) maximum eigenvalue problem, with solution given by $\boldsymbol{C}^* = (1 - \boldsymbol{a}^{*\mathsf{T}}\boldsymbol{a}^*)\boldsymbol{v}_{\max}\boldsymbol{v}_{\max}^\mathsf{T}$, so that $\boldsymbol{D}^* = \boldsymbol{a}^*\boldsymbol{a}^{*\mathsf{T}} + (1 - \boldsymbol{a}^{*\mathsf{T}}\boldsymbol{a}^*)\boldsymbol{v}_{\max}\boldsymbol{v}_{\max}^\mathsf{T}$. This essentially reduces finding $P$ to a 2-dimensional problem, which can be solved in closed form [KNW13, Lemma 4.1]. It is easy to verify that the mixture in the theorem has the desired mean $\boldsymbol{a}^*$ and second moment $\boldsymbol{D}^*$. See Figure 2 for the geometrical intuition.

$\square$

Notice that both the minimax and maximin strategies only depend on $\boldsymbol{W}$ through $\lambda_{\max}$ and $\boldsymbol{v}_{\max}$.

## 5.1 Minimax Analysis of the Ball Game

With the above lemma, we can compute the value and strategies for the ball game in an analogous way to Theorem 4.3. Again, we find that the value function at the end of the game is quadratic in the state, and, surprisingly, remains quadratic under the backwards induction.

**Theorem 5.2.** *Consider the $T$-round ball game with loss $\frac{1}{2}\|\boldsymbol{a} - \boldsymbol{x}\|_{\boldsymbol{W}}^2$. After $n$ rounds, the value-to-go for a state with statistics $\boldsymbol{s} = \sum_{t=1}^n \boldsymbol{x}_t$ and $\sigma^2 = \sum_{t=1}^n \boldsymbol{x}_t^\mathsf{T}\boldsymbol{W}^{-1}\boldsymbol{x}_t$ is*

$$V(\boldsymbol{s}, \sigma^2, n) = \frac{1}{2}\boldsymbol{s}^\mathsf{T}\boldsymbol{A}_n\boldsymbol{s} - \frac{1}{2}\sigma^2 + \gamma_n.$$

*The minimax strategy plays*

$$\boldsymbol{a}^*(\boldsymbol{s}, \sigma^2, n) = \left(\lambda_{\max}\boldsymbol{I} - (\boldsymbol{A}_{n+1} - \boldsymbol{W}^{-1})\right)^{-1}\boldsymbol{A}_{n+1}\boldsymbol{s}$$

*and the maximin strategy plays two unit length vectors with*

$$\Pr\left(\boldsymbol{x} = \boldsymbol{a}_\perp \pm \sqrt{1 - \boldsymbol{a}_\perp^\mathsf{T}\boldsymbol{a}_\perp}\,\boldsymbol{v}_{\max}\right) = \frac{1}{2} \pm \frac{1}{2}\sqrt{\frac{\boldsymbol{a}_\parallel^\mathsf{T}\boldsymbol{a}_\parallel}{1 - \boldsymbol{a}_\perp^\mathsf{T}\boldsymbol{a}_\perp}},$$

*where $\lambda_{\max}$ and $\boldsymbol{v}_{\max}$ correspond to the largest eigenvalue of $\boldsymbol{A}_{n+1}$ and $\boldsymbol{a}_\perp$ and $\boldsymbol{a}_\parallel$ are the components of $\boldsymbol{a}^*$ perpendicular and parallel to $\boldsymbol{v}_{\max}$. The coefficients $\boldsymbol{A}_n$ and $\gamma_n$ are determined recursively by base case $\boldsymbol{A}_T = \frac{1}{T}\boldsymbol{W}^{-1}$ and $\gamma_T = 0$ and recursion*

$$\boldsymbol{A}_n = \boldsymbol{A}_{n+1}\left(\boldsymbol{W}^{-1} + \lambda_{\max}\boldsymbol{I} - \boldsymbol{A}_{n+1}\right)^{-1}\boldsymbol{A}_{n+1} + \boldsymbol{A}_{n+1}$$

$$\gamma_n = \frac{1}{2}\lambda_{\max} + \gamma_{n+1}.$$

*Proof outline.* The proof is by induction on the number $n$ of rounds played. In the base case $n = T$ we find (see (3)) $\boldsymbol{A}_T = \frac{1}{T} \boldsymbol{W}^{-1}$ and $\gamma_T = 0$. For the the induction step, we need to calculate

$$V(\boldsymbol{s}, \sigma^2, n) = \inf_{\boldsymbol{a} \in \bigcirc} \sup_{\boldsymbol{x} \in \bigcirc} \frac{1}{2} \|\boldsymbol{a} - \boldsymbol{x}\|_{\boldsymbol{W}}^2 + V(\boldsymbol{s} + \boldsymbol{x}, \sigma^2 + \boldsymbol{x}^\mathsf{T} \boldsymbol{W}^{-1} \boldsymbol{x}, n + 1).$$

Using the induction hypothesis, we expand the right-hand-side to

$$\inf_{\boldsymbol{a} \in \bigcirc} \sup_{\boldsymbol{x} \in \bigcirc} \frac{1}{2} \|\boldsymbol{a} - \boldsymbol{x}\|_{\boldsymbol{W}}^2 + \frac{1}{2}(\boldsymbol{s} + \boldsymbol{x})^\mathsf{T} \boldsymbol{A}_{n+1}(\boldsymbol{s} + \boldsymbol{x}) - \frac{1}{2}(\sigma^2 + \boldsymbol{x}^\mathsf{T} \boldsymbol{W}^{-1} \boldsymbol{x}) + \gamma_{n+1}.$$

which we can evaluate by applying Lemma 5.1 with $\boldsymbol{A} = \boldsymbol{A}_{n+1} - \boldsymbol{W}^{-1}$ and $\boldsymbol{b} = \boldsymbol{s}^\mathsf{T} \boldsymbol{A}_{n+1}$. Collecting terms and matching with $V(\boldsymbol{s}, \sigma^2, n) = \frac{1}{2} \boldsymbol{s}^\mathsf{T} \boldsymbol{A}_n \boldsymbol{s} - \frac{1}{2} \sigma^2 + \gamma_n$ yields the recursion for $\boldsymbol{A}_n$ and $\gamma_n$ as well as the given minimax and maximin strategies. As before, much of the algebra has been moved to the appendix. $\square$

**Understanding the eigenvalues of $\boldsymbol{A}_n$** As we have seen from the $\boldsymbol{A}_n$ recursion, the eigensystem is always the same as that of $\boldsymbol{W}^{-1}$. Thus, we can characterize the minimax strategy completely by its effect on the eigenvalues of $\boldsymbol{W}^{-1}$. Denote the eigenvalues of $\boldsymbol{A}_n$ and $\boldsymbol{W}^{-1}$ to be $\lambda_n^i$ and $\nu_i$, respectively, with $\lambda_{n-1}^1$ corresponding to the largest eigenvalue. The eigenvalues follow:

$$\lambda_{n-1}^i = \frac{(\lambda_n^i)^2}{\nu_i + \lambda_n^1 - \lambda_n^i} + \lambda_i = \frac{\lambda_n^i(\nu_i + \lambda_n^1)}{\nu_i + \lambda_n^1 - \lambda_n^i},$$

which leaves the order of $\lambda_n^i$ unchanged. The largest eigenvalue $\lambda_n^1$ satisfies the recurrence $\lambda_T^1/\nu_1 = 1/T$ and $\lambda_n^1/\nu_1 = \left(\lambda_{n+1}^1/\nu_1\right)^2 + \lambda_{n+1}^1/\nu_1$, which, remarkably, is the same recurrence for the $\alpha_n$ parameter in the Brier game, i.e. $\lambda_n^{\max} = \alpha_n \nu_{\max}$.

This observation is the key to analyzing the minimax regret.

**Theorem 5.3.** *The minimax regret of the $T$-round ball game satisfies*

$$V \leq \frac{1 + \ln(T)}{2} \lambda_{\max}(\boldsymbol{W}^{-1}).$$

*Proof.* We have $V = V(\boldsymbol{0}, 0, 0) = \gamma_0 = \sum_{n=1}^T \lambda_n^{\max} = \lambda_{\max}(\boldsymbol{W}^{-1}) \sum_{n=1}^T \alpha_n$, the last equality following from the discussion above. The proof of Theorem 4.4 gives the bound on $\sum_{n=1}^T \alpha_n$. $\square$

Taking stock, we find that the minimax regrets of the Brier game (Theorems 4.3) and ball game (Theorems 5.2) have identical dependence on the horizon $T$ but differ in a complexity factor arising from the interaction of the action space and the loss matrix $\boldsymbol{W}$.

## 6 Conclusion

In this paper, we have presented two games that, unexpectedly, have computationally efficient minimax strategies. While the structure of the square Mahalanobis distance is important, it is the interplay between the loss and the constraint set that allows efficient calculation of the backwards induction, value-to-go, and achieving strategies. For example, the square Mahalanobis game with $\ell_1$ ball action spaces does not admit a quadratic value-to-go unless $\boldsymbol{W} = \boldsymbol{I}$.

We emphasize the low computational cost of this method despite the exponential blow-up in state space size. In the Brier game, the $\alpha_n$ coefficients need to be precomputed, which can be done in $O(T)$ time. Similarly, computation of the eigenvalues of the $\boldsymbol{A}_n$ coefficients for the ball game can be done in $O(TK + K^3)$ time. Then, at each iteration of the algorithm, only matrix-vector multiplications between the current state and the precomputed parameters are required. Hence, playing either $T$ round game requires $O(TK^2)$ time. Unfortunately, as is the case with most minimax algorithms, the time horizon must be known in advance.

There are many different future directions. We are currently pursuing a characterization of action spaces that permit quadratic value functions under squared Mahalanobis loss, and investigating connections between losses and families of value functions closed under backwards induction. There is some notion of conjugacy between losses, value-to-go functions, and action spaces, but a generalization seems difficult: the Brier game and ball game worked out for seemingly very different reasons.

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
