[Supplementary Material]

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

} C\boldsymbol{d} - 1}{\mathbf{1}^\mathsf{T} C\mathbf{1}}\mathbf{1} \right) \;=\; \left( C - \frac{C\mathbf{1}\mathbf{1}^\mathsf{T} C}{\mathbf{1}^\mathsf{T} C\mathbf{1}} \right)\boldsymbol{d} + \frac{C\mathbf{1}}{\mathbf{1}^\mathsf{T} C\mathbf{1}}$$

*provided that $\boldsymbol{p}^*$ is in the simplex.*

*Proof.* We solve for the optimal $\boldsymbol{p}^*$. Introducing Lagrange multiplier $\lambda$ for the constraint $\sum_k p_k = 1$, we need to have $\boldsymbol{p} = \boldsymbol{C}\,(\boldsymbol{d} - \lambda\mathbf{1})$ which results in $\lambda = \frac{\mathbf{1}^\intercal \boldsymbol{C}\boldsymbol{d} - 1}{\mathbf{1}^\intercal \boldsymbol{C}\mathbf{1}}$. Thus, the maximizer equals $\boldsymbol{p}^* = \boldsymbol{C}\left(\boldsymbol{d} - \frac{\mathbf{1}^\intercal \boldsymbol{C}\boldsymbol{d} - 1}{\mathbf{1}^\intercal \boldsymbol{C}\mathbf{1}}\mathbf{1}\right)$ which produces objective value $\frac{1}{2}\left(\boldsymbol{d} + \frac{\mathbf{1}^\intercal \boldsymbol{C}\boldsymbol{d} - 1}{\mathbf{1}^\intercal \boldsymbol{C}\mathbf{1}}\mathbf{1}\right)^\intercal \boldsymbol{C}\left(\boldsymbol{d} - \frac{\mathbf{1}^\intercal \boldsymbol{C}\boldsymbol{d} - 1}{\mathbf{1}^\intercal \boldsymbol{C}\mathbf{1}}\mathbf{1}\right)$. The statement follows from simplification. $\square$

This lemma allows us to compute the value and saddle point whenever the future payoff is quadratic.

**Lemma 4.2.** *Fix symmetric matrices $\boldsymbol{W} \succeq \boldsymbol{0}$ and $\boldsymbol{A}$ such that $\boldsymbol{W}^{-1} + \boldsymbol{A} \succeq \boldsymbol{0}$, and a vector $\boldsymbol{b}$. The optimization problem*

$$\min_{\boldsymbol{a}\in\triangle} \max_{\boldsymbol{x}\in\triangle} \frac{1}{2}\|\boldsymbol{a} - \boldsymbol{x}\|_{\boldsymbol{W}}^2 + \frac{1}{2}\boldsymbol{x}^\intercal \boldsymbol{A}\boldsymbol{x} + \boldsymbol{b}^\intercal \boldsymbol{x}$$

*achieves its value*

$$\frac{1}{2}\boldsymbol{c}^\intercal \boldsymbol{W}\boldsymbol{c} - \frac{1}{2}\frac{(\mathbf{1}^\intercal \boldsymbol{W}\boldsymbol{c} - 1)^2}{\mathbf{1}^\intercal \boldsymbol{W}\mathbf{1}} \qquad \textit{

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

## A  Proofs

*Proof of Theorem 4.3.* We prove by induction. Recall that $V(s, \sigma^2, T)$, the value at the end of the game, already has

$$V(s, \sigma^2, T) = \frac{1}{2T} s^\mathsf{T} W^{-1} s - \frac{1}{2}\sigma^2,$$

corresponding to $\alpha_T = \frac{1}{T}$ and $\gamma_T = 0$. Adding a constant to the objective in Lemma 4.2 only adds the same offset to the value without changing the saddle point. Now, assume the induction hypothesis for $n \leq T$ rounds. Using the abbreviation $d = \operatorname{diag}(W^{-1})$, we then need to evaluate

$$
\begin{aligned}
V(s, \sigma^2, n-1) &= \min_{a \in \triangle} \max_{x \in \triangle} \frac{1}{2} \|a - x\|_W^2 + \frac{1}{2}\alpha_n (s+x)^\mathsf{T} W^{-1}(s+x) \\
&\quad - \frac{1}{2}(\sigma^2 + x^\mathsf{T} W^{-1} x) + \frac{1}{2}(1 - n\alpha_n) d^\mathsf{T}(s+x) + \gamma_n. \\
&= \min_{a \in \triangle} \max_{x \in \triangle} \frac{1}{2} \|a - x\|_W^2 + \frac{1}{2}\alpha_n x^\mathsf{T} W^{-1} x - \frac{1}{2} x^\mathsf{T} W^{-1} x \\
&\quad + \frac{1}{2}(1 - n\alpha_n) d^\mathsf{T} x + \alpha_n s^\mathsf{T} W^{-1} x + \frac{1}{2}\alpha_n s^\mathsf{T} W^{-1} s \\
&\quad + \frac{1}{2}(1 - n\alpha_n) d^\mathsf{T} s - \frac{1}{2}\sigma^2 + \gamma_n.
\end{aligned}
$$

Applying Lemma 4.2 with $A = (\alpha_n - 1)W^{-1}$ and $b = \frac{1}{2}(1 - n\alpha_n) d + \alpha_n W^{-1} s$ produces the value (where all time indices are $n$)

$$
\begin{aligned}
V(s, \sigma^2, n-1) &= \frac{1}{2}(\alpha^2 + \alpha) s^\mathsf{T} W^{-1} s + \left(\alpha \frac{1 - (n-1)\alpha}{2} + \frac{1}{2}(1 - n\alpha)\right) d^\mathsf{T} s \\
&\quad + \frac{1}{2}\left(\frac{1 - (n-1)\alpha}{2}\right)^2 d^\mathsf{T} W d \\
&\quad - \frac{1}{2} \frac{\left(\frac{1-(n-1)\alpha}{2} \mathbf{1}^\mathsf{T} W d + \alpha \mathbf{1}^\mathsf{T} s - 1\right)^2}{\mathbf{1}^\mathsf{T} W \mathbf{1}} \\
&\quad - \frac{1}{2}\sigma^2 + \gamma
\end{aligned}
$$

and the strategy

$$p^* = \left(W - \frac{W \mathbf{1}\mathbf{1}^\mathsf{T} W}{\mathbf{1}^\mathsf{T} W \mathbf{1}}\right)\left(\frac{1}{2}(1 - (n-1)\alpha_n) d + \alpha_n W^{-1} s\right) + \frac{W \mathbf{1}}{\mathbf{1}^\mathsf{T} W \mathbf{1}}.$$

Using the fact that $\mathbf{1}^\mathsf{T} s = n - 1$, the value can be further simplified to

$$
\begin{aligned}
V(s, \sigma^2, n-1) &= \frac{1}{2}(\alpha^2 + \alpha) s^\mathsf{T} W^{-1} s + \frac{1}{2}\left(1 - (n-1)(\alpha^2 + \alpha)\right) d^\mathsf{T} s \\
&\quad + (1 - (n-1)\alpha)^2 \left(\frac{1}{8} d^\mathsf{T} W d - \frac{1}{2}\frac{\left(\frac{1}{2}\mathbf{1}^\mathsf{T} W d - 1\right)^2}{\mathbf{1}^\mathsf{T} W \mathbf{1}}\right) - \frac{1}{2}\sigma^2 + \gamma.
\end{aligned}
$$

This establishes the induction step upon setting

$$
\begin{aligned}
\alpha_n &= \alpha_{n+1}^2 + \alpha_{n+1}, \\
\gamma_n &= \gamma_{n+1} + (1 - n\alpha_{n+1})^2 \left(\frac{1}{8} d^\mathsf{T} W d - \frac{1}{2}\frac{\left(\frac{1}{2}\mathbf{1}^\mathsf{T} W d - 1\right)^2}{\mathbf{1}^\mathsf{T} W \mathbf{1}}\right).
\end{aligned}
$$

Finally, we need to verify that the strategies above respect the constraints of the game, i.e. that $a^*$ and $p^*$ are in the simplex. Otherwise, the above calculations do not correspond to the minimax strategy. We need to show for all $s \geq 0$ with sum $n - 1$ that

$$\alpha_n s + \frac{1}{2}(1 - (n-1)\alpha_n) W d \geq \left(\alpha_n(n-1) + \frac{1}{2}(1 - (n-1)\alpha_n)(\mathbf{1}^\mathsf{T} W d) - 1\right) \frac{W \mathbf{1}}{\mathbf{1}^\mathsf{T} W \mathbf{1}}.$$

As this needs to hold for all $s \geq 0$, we need in fact that component-wise

$$\frac{1}{2}\left(1 - (n-1)\alpha_n\right)\boldsymbol{Wd} \geq \left(\alpha_n(n-1) + \frac{1}{2}\left(1 - (n-1)\alpha_n\right)(\mathbf{1}^\mathsf{T}\boldsymbol{Wd}) - 1\right)\frac{\boldsymbol{W}\mathbf{1}}{\mathbf{1}^\mathsf{T}\boldsymbol{W}\mathbf{1}},$$

that is

$$\left(\boldsymbol{W} - \frac{\boldsymbol{W}\mathbf{1}\mathbf{1}^\mathsf{T}\boldsymbol{W}}{\mathbf{1}^\mathsf{T}\boldsymbol{W}\mathbf{1}}\right)\operatorname{diag}(\boldsymbol{W}^{-1}) \succeq \frac{2\left((n-1)\alpha_n - 1\right)}{1 - (n-1)\alpha_n}\frac{\boldsymbol{W}\mathbf{1}}{\mathbf{1}^\mathsf{T}\boldsymbol{W}\mathbf{1}} = -2\frac{\boldsymbol{W}\mathbf{1}}{\mathbf{1}^\mathsf{T}\boldsymbol{W}\mathbf{1}}.$$

$\square$

*Proof of Theorem 5.2.* Recall that we needed to calculate

$$V(\boldsymbol{s}, \sigma^2, n) = \inf_{\boldsymbol{a}}\sup_{\boldsymbol{x}}\frac{1}{2}\|\boldsymbol{a} - \boldsymbol{x}\|_{\boldsymbol{W}}^2 + \frac{1}{2}(\boldsymbol{s} + \boldsymbol{x})^\mathsf{T}\boldsymbol{A}_{n+1}(\boldsymbol{s} + \boldsymbol{x}) - \frac{1}{2}(\sigma^2 + \boldsymbol{x}^\mathsf{T}\boldsymbol{W}^{-1}\boldsymbol{x}) + \gamma_{n+1}$$

which may be reorganized to

$$\frac{1}{2}\boldsymbol{s}^\mathsf{T}\boldsymbol{A}_{n+1}\boldsymbol{s} - \frac{1}{2}\sigma^2 + \gamma_{n+1} + \inf_{\boldsymbol{a}}\sup_{\boldsymbol{x}}\frac{1}{2}\|\boldsymbol{a} - \boldsymbol{x}\|_{\boldsymbol{W}}^2 + \frac{1}{2}\boldsymbol{x}^\mathsf{T}\left(\boldsymbol{A}_{n+1} - \boldsymbol{W}^{-1}\right)\boldsymbol{x} + \boldsymbol{s}^\mathsf{T}\boldsymbol{A}_{n+1}\boldsymbol{x}.$$

We now apply Lemma 5.1 with $\boldsymbol{A} = \boldsymbol{A}_{n+1} - \boldsymbol{W}^{-1}$ and $\boldsymbol{b} = \boldsymbol{s}^\mathsf{T}\boldsymbol{A}_{n+1}$. Let $\lambda_{\max}$ be the largest eigenvalue of $\boldsymbol{A}_{n+1} - \boldsymbol{W}^{-1} + \boldsymbol{W}^{-1} = \boldsymbol{A}_{n+1}$. If we have

$$1 \geq \boldsymbol{s}^\mathsf{T}\boldsymbol{A}_{n+1}\left(\boldsymbol{W}^{-1} + \lambda_{\max}\boldsymbol{I} - \boldsymbol{A}_{n+1}\right)^{-2}\boldsymbol{A}_{n+1}\boldsymbol{s}, \tag{5}$$

and then the value equals

$$\frac{1}{2}\lambda_{\max} + \frac{1}{2}\boldsymbol{s}^\mathsf{T}\boldsymbol{A}_{n+1}\left(\boldsymbol{W}^{-1} + \lambda_{\max}\boldsymbol{I} - \boldsymbol{A}_{n+1}\right)^{-1}\boldsymbol{A}_{n+1}\boldsymbol{s}.$$

Writing this out as a quadratic in $\boldsymbol{s}$ we get

$$\frac{1}{2}\boldsymbol{s}^\mathsf{T}\boldsymbol{A}_{n+1}\left(\boldsymbol{W}^{-1} + \lambda_{\max}\boldsymbol{I} - \boldsymbol{A}_{n+1}\right)^{-1}\boldsymbol{A}_{n+1}\boldsymbol{s} + \frac{1}{2}\lambda_{\max},$$

so if we assert that $V(\boldsymbol{s}, \sigma^2, n) = \frac{1}{2}\boldsymbol{s}^\mathsf{T}\boldsymbol{A}_n\boldsymbol{s} - \frac{1}{2}\sigma^2 + \gamma_n$, we find that we must have the correspondence:

$$\boldsymbol{A}_n = \boldsymbol{A}_{n+1}\left(\boldsymbol{W}^{-1} + \lambda_{n+1}^{\max}\boldsymbol{I} - \boldsymbol{A}_{n+1}\right)^{-1}\boldsymbol{A}_{n+1} + \boldsymbol{A}_{n+1},$$

$$\gamma_n = \frac{1}{2}\lambda_{n+1}^{\max} + \gamma_{n+1}.$$

The final piece is to check Equation (5). To see this, let $\boldsymbol{\nu}$ denote the eigenvalues of $\boldsymbol{W}^{-1}$. Observe that the largest eigenvalue of $\boldsymbol{A}_{n+1}\left(\boldsymbol{W}^{-1} + \lambda_{\max}\boldsymbol{I} - \boldsymbol{A}_{n+1}\right)^{-1}$ is $\lambda_{n+1}^{\max}/\nu_{\max}$ as the mapping $\lambda \mapsto \frac{\lambda}{\nu + \lambda^{\max} - \lambda}$ is monotonic in $\lambda \leq \lambda^{\max}$. The proof now follows by combining the recurrence for the largest eigenvalue from Section 5.1 with the bound on $\alpha_n$ from the proof of Theorem 4.4.

$\square$