[Reviews · NeurIPS 2014]

Submitted by Assigned_Reviewer_7

This paper shows efficient minimax strategies when the loss function is the squared Mahalanobis distance, for two different settings: 1) action and outcome spaces are the probability simplex, and 2) they are the Euclidean ball.

The paper is clearly written with a consistent set of notations. The games considered appear similar to previous work, such as TW00 and ABRT08. The only other direction of proposed novelty is in the choice of the Mahalanobis distance, a generalization of squared euclidean distance.

But here I think a simple transformation that would take the Mahalanobis game, and covert it the vanilla squared euclidean distance version. If you let M = W^{1/2), then we can simply solve the game assuming squared euclidean distance, converting the points a_t to a_t' = M * a_t. We can learn points x_t' in the transformed space, and convert back to the original space by letting x_t = M^{-1} * x_t'. Notice that I'm using the fact that ||a_t' - x_t'||^2 = ||M * a_t - M * x_t||^2 = ||a_t - x_t||_W^2. Unless I am missing something, it appears to me that this simple linear transformation makes the Mahalanobis game no harder than the euclidean game.

The authors also missed several pieces of literature that might have been relevant, as the last few years has seen a burst of such results. These include Mcmahan/Streeter, Abernethy/Warmuth, Mcmahan/Orabona, Rakhlin/Sridharan. Finally, as the authors admit, their proof techniques don't seem to easily generalize beyond the two canonical settings.

* After reading authors' response, I changed my rating. The games considered in this paper (and the resulting minimax strategies) indeed are distinct from the previous work. The proofs seem sound, with the authors' correction and extra explanation.
Summary: This paper presents efficient minimax strategies for a challenging problem, but the results are limited in scope and not novel.

Submitted by Assigned_Reviewer_17

The problem of online prediction with quadratic distance (Mahalanobis distance) between the predicted vector and outcome as loss is considered. For two specific problems, one where prediction and outcome sets are the simplex and the other case when the two sets are l2 balls exact minimax strategies are provided and these are shown to even be computationally efficient. Regret bound for these games that have logarithmic dependence on horizon T are also proven.

The paper is well written and easy to follow. The one drawback I would think is that for both these games exact minimax analysis seem to come out of the exact form of the games and so it doesn’t seem like the technique sheds light on possible approximate algorithms one could use for quadratic games over more general sets.

Some further comments/questions :

1. While the Azoury-Warmuth-Vovk algorithm is for a different problem of linear regression, it seems like there are striking similarities between the analysis here and the analysis for that algorithm (esp. the one that shows that it is the minimax algorithm). Given this similarity I am curious if one can consider the quadratic game in $d$ dimensional space but with no a priori restriction on the sets. Of course the regret bound will be dependent of the sequence. For instance in the Azoury-Warmuth-Vovk algorithm the regret bound is in terms of max value of outcome variable. Similarly perhaps one can get a regret bound interns of max l2 norm of outcome vector (in hind sight).

2. I am curious as to what kind of mahalanobis balls can the euclidean sets proof be pushed. That is assume the prediction and outcome sets are l2 balls w.r.t. matrix H. What kind of H can minimax algorithm be provided. For instance what if H is rank one, Or if H is invertible?
Summary: Overall I find this paper interesting. I do feel that to extend its applicability to general problems, working on perhaps approximate methods (rather than minimax but till rising from minimax type techniques) for more general sets would go a long way.

Submitted by Assigned_Reviewer_42

Summary:

This paper derived the minimax solutions for online prediction problems where the loss between the prediction and the outcome is measured by the squared Euclidean distance and the squared Mahalanobis distance. Two cases of prediction and action space are included: probability simplex and L_2 ball. For each problem, it provides detailed analysis for the optimal strategy and regret bound. This is a purely theoretical work without experiments.

Quality:

This paper is of poor quality. Although lots of detailed analysis are given, there seems an important mistake in its mathematic calculation, making the correctness of one of the two contributions of this paper, the minimax strategy for probability simplex, being in doubt.

Clarity:

This presentation is clear in most cases, although there are a few typos. Another problem appears at the end of the first page. To make the presentation clear, it should provide a detailed mathematic definition for the probability simplex.

Originality:

This seems an original work. Although minimax analysis has been proposed in existing works, the specific problem, Mahalanobis distance loss in simplex or L2 ball, has not been studied.

Significance:

This is obviously a theoretical work. There is no experiment in the paper. Also there is no discussion for its application to real world problems. It is unknown for the significance of the theory.

Comments on the main problem, the simplex problem:

One of the two main contributions of this work is the minimax solutions for predictions and actions on probability simplex. And all this section is based on Lemma 4.1. This lemma is about the solution of an optimization problem on “probability simplex”. Since there is no clear definition of “probability simplex” in this paper, we have to use the usual definition: sum_i(p_i)=1 and p_i > 0. But in this lemma, “positive” is not mentioned as a constraint and not involved in Lagrange multiplier. Thus, the optimizer p* and optimized value are obtained without positive constraint. In addition, it is easy to show that the p* derived in this lemma is not always positive: for example, take c=[2,1;1,2] which is symmetric and positive definite, d=[2;5], p*=c*(d-(sum(c*d)-1)/sum(sum(c)))=[-1;2], not positive.

Theorem 4.2 is based on Lemma 4.1, where p is assumed as the distribution of k and thus should be positive. This obviously contradicts with the assumption (p does not need to be positive) of lemma 4.1. Therefore, it is not proper for Theorem 4.2 to apply the conclusion of Lemma 4.1. We can conclude that Theorem 4.2 is wrong.

Theorem 4.3 and Lemma 4.4 are based on 4.2 and thus should be incorrect. Thus, for the simplex problem, both the strategy and the regret bound are problematic.
Summary: The paper proposed minimax strategies for addressing online prediction problems where the loss between the prediction and the outcome is measured by the squared Euclidean distance and the generalized squared Mahalanobis distance. The paper seems to have some flaw in the formulations, which must be fixed if the paper would be accepted. Besides, it is unknown for the practical values of the theoretical study. I think the paper presentation has to be greatly improved if the paper is to be accepted.
Author Feedback
Author rebuttal: Reviewer 17:

Thanks for your kind words and interesting questions.

Question 1:

Relaxing the assumptions on the outcomes and obtaining data dependent regret bounds is certainly a worthwhile endeavour, and we share your curiosity. We do not have any results in this direction as of yet, but thanks for the pointer.

Question 2: there is (unfortunately) no generality gained by considering the game with Mahalanobis loss over a Mahalanobis ball. The reason for this is (see our response to Reviewer 7 below) that this game is equivalent to a game with Mahalanobis loss on the Euclidean ball.

In the paper we purposefully avoided the pathological singular Mahalanobis metrics. In such metrics the norm ball is unbounded and the minimax regret is infinite (already in 1 round). It is quite likely that something interesting can be still be said in the form of a data dependent bound, see Question 1

Reviewer 42:

The reviewer argues against publication at NIPS based on a flaw in the mathematical details. The reviewer is correct in identifying the importance of the necessity of the optimizer p*'s remaining in the simplex as the algorithm is not well defined otherwise. Fortunately, we present the necessary and sufficient condition for the maximin strategy to remain in the simplex in equation (4). The proof that the maximin strategy remains in the simplex can be found in Appendix A, lines 526-544.

The statement of Lemma 4.1 was indeed incorrect. Analogous to Theorem 4.2, the lemma should read: "the problem ... has value ... and is attained at optimizer p = ... provided that p >= 0".

The structure of the argument and importance of equation (4) were not properly emphasised. We can easily solve this problem by adding the following paragraph (at line 149) to clarify the structure of the argument to come:

"The structure of the paper is as follows. In Lemma 4.1 and Theorem 4.2 the conclusions (value and optimisers) are obtained under the proviso that the given optimizer lies in the simplex. In our main result Theorem 4.3 we apply these auxiliary results to our minimax analysis and argue that the maximiser indeed lies in the simplex. This fact is a consequence of the alignment assumption (4) on the Mahalanobis matrix W. The full details of this step are in Appendix A."

We thank the reviewer for pointing out the mistake in the lemma statement, and kindly ask him or her to reconfirm that the result is
indeed correct, and to consequently adjust the assessment of the quality of our paper.

Reviewer 7:

The reviewer's idea to reduce the Mahalanobis game to the vanilla squared loss game is natural and indeed something we considered. As the reviewer correctly observes, this game with Mahalanobis loss over the Euclidean ball is equivalent to the game with Euclidean loss over the inverse Mahalanobis ball. However, either game is fundamentally different from the Euclidean loss game over the Euclidean ball; in the Mahalanobis case, as we show, the minimax and maximin strategies do depend non-trivially on the Mahalanobis coefficients W.

The results in [TW00] only apply to the game with Euclidean loss over the Euclidean ball and our results are more general. Indeed, the algorithm we find has each eigenvalue of the $A_{n,r}$ matrix (an essential ingredient of the minimax algorithm) decay differently, whereas in [TW00], they start out and decay identically.

Finally, we'd like to thank the reviewer for their ideas and references to relevant literature.